# Programmed Cell Death in the Small Intestine: Implications for the Pathogenesis of Celiac Disease

**DOI:** 10.3390/ijms22147426

**Published:** 2021-07-10

**Authors:** Federico Perez, Carolina Nayme Ruera, Emanuel Miculan, Paula Carasi, Fernando Gabriel Chirdo

**Affiliations:** Instituto de Estudios Inmunológicos y Fisiopatológicos (IIFP), UNLP, CONICET, CIC PBA, Departamento de Ciencias Biológicas, Facultad de Ciencias Exactas, Universidad Nacional de La Plata, La Plata B1900, Argentina; nayme.cr@gmail.com (C.N.R.); emanuelmiculan@gmail.com (E.M.); paulacarasi@gmail.com (P.C.)

**Keywords:** celiac disease, alarmins, programmed cell death, inflammation, small intestine

## Abstract

The small intestine has a high rate of cell turnover under homeostatic conditions, and this increases further in response to infection or damage. Epithelial cells mostly die by apoptosis, but recent studies indicate that this may also involve pro-inflammatory pathways of programmed cell death, such as pyroptosis and necroptosis. Celiac disease (CD), the most prevalent immune-based enteropathy, is caused by loss of oral tolerance to peptides derived from wheat, rye, and barley in genetically predisposed individuals. Although cytotoxic cells and gluten-specific CD4^+^ Th1 cells are the central players in the pathology, inflammatory pathways induced by cell death may participate in driving and sustaining the disease through the release of alarmins. In this review, we summarize the recent literature addressing the role of programmed cell death pathways in the small intestine, describing how these mechanisms may contribute to CD and discussing their potential implications.

## 1. Introduction

### 1.1. Programmed Cell Death (PCD)

Many new cell death pathways have been discovered in recent years. From the original concept of apoptosis as a unique form of immunologically silent programmed cell death, to proinflammatory necrosis, a broad spectrum of different pathways is known to develop in specific conditions. Apoptosis is characterized by cell shrinkage and chromatin condensation, followed by fragmentation of the entire cell into small apoptotic bodies, which are cleared away by macrophages without initiating an inflammatory response. Unlike apoptosis, other forms of PCD are not immunologically silent and are involved in driving and maintaining a variety of metabolic and inflammatory disorders. These pathways include necroptosis, pyroptosis and ferroptosis and they can result in the release of proinflammatory molecules such as alarmins (IL-33, HMGB1, IL-1α) and proinflammatory cytokines (IL-1β and IL-18) [1,2,3]. Importantly, inflammatory PCD enables the release of molecules such as IL-1β without cell death occurring [4]. Studying the molecular pathways involved in these processes is important in order to gain insight into the pathogenesis of inflammatory disorders and for the development of therapeutic interventions.

### 1.2. Celiac Disease: A Complex Small Intestine Pathology

#### 1.2.1. Antigen-Specific CD4^+^ T Cells

Celiac disease (CD) is a highly prevalent chronic inflammatory enteropathy which occurs in genetically susceptible individuals as a consequence of an immune response to gluten proteins derived from wheat, barley and rye [5,6]. It affects the proximal small intestine, leading to villus atrophy and crypt hyperplasia, together with increased numbers of lamina propria and intraepithelial lymphocytes (IELs). However, the histological changes and clinical presentation are variable, and many cases remain asymptomatic, leading to a large number of undiagnosed patients [7]. The histological changes are caused by an immune response to dietary gluten in the small intestine mucosa. As gluten peptides are resistant to proteolysis by gastrointestinal enzymes, long peptides remain in the lumen of the intestine. After crossing the epithelial layer [8], some of these peptides are deamidated by the enzymatic activity of transglutaminase 2 (TG2), generating epitopes with the ability to bind to the disease specifying MHC molecules HLA-DQ2 and DQ8. In the lamina propria, native and deamidated gluten derived peptides are taken up and presented in HLA molecules by dendritic cells (DC). Based on studies in mice, it is believed that lamina propria DC migrate to the draining mesenteric lymph nodes where they encounter naïve, antigen-specific CD4^+^ T cells and induce their differentiation into Th1 CD4^+^ T cells, which migrate back to the lamina propria [9]. However, this has not been shown directly in humans with CD. In the lamina propria, gluten-specific CD4^+^ cells produce IFNγ, the dominant cytokine in the chronic inflammatory process [6,10,11].

#### 1.2.2. Cytotoxic Mechanisms

Several gluten peptides have been found that can bind to the class I HLA molecules: HLA-A2 and HLA-A*0101 and B*0801 and HLA-A2^+^ CD patients have an increased number of specific gluten-specific, CD8^+^ cytotoxic T lymphocytes (CTLs) in lamina propria which express IFNγ, CD95 and granzyme B (GZMB) when stimulated with gliadin [12,13,14]. However, it should be noted that there is little direct association between CD and specific HLA class I alleles [15], suggesting that antigen-specific CTL activity is not part of the genetic susceptibility to CD.

A hallmark of active CD is an increased number of IELs in the small intestine, even when pathology is not severe [16]. These IELs comprise cytotoxic CD8^+^ T lymphocytes (CTL), γ/δ T cells and NK cells. Similar cells are found in the lamina propria of the small intestine of CD patients [11,13,17] and the mucosa of untreated CD contains high levels of cytokines that can potentiate cytotoxic lymphocyte activity such as IL-15, IL-21, type I and II Interferons (IFNs) [18,19,20,21].

IELs constitute the largest T cell compartment in the body with one IEL being found for every 10 epithelial cells in the human small intestine [16,22]. Two main populations of IEL are present, adaptive (or conventional) and innate-like or unconventional IELs [23]. Adaptive IELs are TCRαβ^+^ T cells with a memory phenotype and consist of CD8αβ^+^ (~80%) and CD4^+^ (~10%) subsets, all of which are derived from naïve T cells that have been primed by specific peptide antigens presented by MHC on DC in secondary lymphoid organs. In contrast, innate-like IELs are activated by cytokines (IL-15) and NK receptors, in a TCR-independent manner, and can represent 5–30% of human IELs [22]. Innate-like IELs include both TCRαβ^+^ CD8αα^+^ and TCRγ/δ^+^ T cells. A hallmark of untreated CD is an expanded population of TCRγ/δ^+^ IELs that recognise the non-classical class I MHC molecules, butyrophilin-like (BTNL) molecules BTNL3/8, and whose numbers remain high even after long period on a gluten-free diet (GFD) [24].

As with NK cells, innate-like IELs can be activated in CD via recognition of non-classical class I MHC molecules such as MIC-A on stressed epithelial cells by the NK cell receptor NKG2D on IELs [24,25,26,27,28]. NKG2C/CD94 is a further NK cell receptor, which interacts with the non-classical class I MHC molecule HLA-E present on epithelial cells exposed to IFNs and other inflammatory cytokines [29,30]. The NKG2C/CD94-HLA-E interaction is thought to induce IFNγ production by CTLs and enhance their cytotoxic activity in CD patients [17,31,32]. Type I IFNs may also potentiate the non-antigen-specific cytotoxic activity of CTLs against epithelial cells [21].

#### 1.2.3. Innate Immunity

In addition to antigen-specific CD4^+^ T cell mediated immunity, several other aspects of the immune response are involved in CD. Recent studies have underlined the critical importance of external factors triggering innate immunity in the pathogenesis of CD. These include viral infections [9], dysbiosis of the microbiota [33], amylase-trypsin inhibitors [34] and a non-T cell epitope peptide of gliadin (known as p31-43) [35]. Innate immunity and chronic inflammation have a direct impact on potentiating the activation of both gluten-specific CD4^+^ T cell and IELs in CD [9,24,36].

Several proinflammatory effects have been described for p31-43 peptide in vitro and in vivo [35]. It does not bind to the HLA-DQ2 or DQ8 molecules [37], and a specific surface receptor has not been identified [38]. However, it has been shown that p31-43, binds to the nucleotide-binding domain 1 (NBD1) subunit of the cystic fibrosis transmembrane conductance regulator (CFTR), reducing its ATPase activity and causing an increase in reactive oxygen species (ROS) generation and a persistent activation of TG2, leading to increased nuclear translocation of NFκB. Activation of the NFκB pathway induces the transcription of pro-inflammatory cytokines such as IL-17A, IL-21 and IL-15, together with production of active IL-1β [39]. Some of these effects may reflect the fact that p31-43 has a poly-proline II structure and forms oligomers stable in solution with pH ranging from 4 to 8 [40,41], resulting in endoplasmic reticulum-stress (ER-stress), the production of ROS, release of proinflammatory mediators such as IL-1β, and the induction of cell death [35,42].

## 2. Apoptosis and Cell Shedding from the Epithelium

Under homeostatic conditions, the epithelial layer of the human small intestine is almost completely renewed every week, with a balance between extrusion of effete enterocytes at the tip of the villi and the production of new cells in the transit-amplifying zone of the crypts [43]. As in other epithelial tissues, the loss of enterocytes from the crowded villus tip is triggered by the stretch-activated channel protein Piezo 1 [44] which leads to the release of sphingosine-1-phosphate (S1P) that drives the release of the cells from integrin-dependent anchoring to the extracellular matrix. This leads to a form of apoptotic cell death known as anoikis and extrusion of cells from the epithelium [44,45]. Inflammatory mediators such as TNFα and IFNγ, as well as microbial associated molecular patterns (MAMPs) and pathogens may increase the speed of this process.

Although this mechanism has not been assessed in CD patients, the presence of high concentrations of IFNγ in celiac mucosa is likely to result in the loosening of the tight junctions between epithelial cells and defective closing over of the extrusion area left by shedding of dying cells. In support of this idea, the junctional protein E-cadherin is essential for closing these exposed spots by elongating neighboring cells, which avoids the formation of transient epithelial gaps [46,47] and its expression is downregulated in epithelial cells exposed to IFNγ and TNFα in vitro [48], as well as in CD [49] and IBD in vivo [44,50].

Apoptosis can occur by intrinsic and extrinsic pathways which trigger a common executioner mechanism (summarized in Figure 1).

### 2.1. Intrinsic Apoptosis

The intrinsic pathway involves permeabilization of the mitochondria outer membrane (MOMP) due to activation of the pore forming proteins BAK (BCL2 antagonist/killer 1) and BAX (BCL2 associated X, apoptosis regulator). This pore facilitates the release of cytochrome-c that binds and activates apoptotic peptidase activating factor (APAF-1) and the initiator caspase-9, together forming the apoptosome which promotes activation of the executioner caspase-3. Caspase-3 activates other procaspases (e.g., caspase-2, -6, -8, and -10), creating an apoptosis-amplifying cascade, which ends in the alteration of the nuclear membrane, the cleavage of intracellular proteins (e.g., PARP), membrane blebbing, and the breakdown of genomic DNA into nucleosomal structures. Release of SMAC (second mitochondria-derived activator of caspases) and the HtrA serine protease 2 (HTRA2) from the mitochondrial membrane also leads to inhibition of the caspase inhibitor X-linked inhibitor of apoptosis protein (XIAP), thus enhancing and sustaining activation of caspase-3. The mitochondrial events in apoptosis are regulated by a balance between pro- and anti-apoptotic proteins of the BCL family. The “BH3-only” domain, t-BID, BIM, NOXA, and PUMA are pro-apoptotic, as they interact with BAK and BAX and promote their membrane permeabilizing actions. On the other hand, proteins belonging to BCL-2 family (BCL-2, BCL-XL, BCL-W, BFL1, MCL1) inhibit apoptosis by competing with the binding of the BH3-only family members to BAX and BAK [51,52].

The intrinsic pathway is triggered by factors such as damage to genomic or mitochondrial DNA, signals derived from damaged organelles (e.g., ER or mitochondrial stress), the inhibition or reduction of growth factors, and the inhibition of intracellular signaling pathways [51]. One of the core mediators of intrinsic apoptosis is the p53 protein, which is activated upon DNA damage induced by oxidants, alkylating agents, or radiation, and it promotes the expression of the pro-apoptotic factors BAX, PUMA and NOXA, while downregulating the expression of anti-apoptotic BCL-2.

One of the most potent immunological triggers for intrinsic apoptosis is the lethal hit delivered by cytotoxic T lymphocytes and NK cells. These cells release granules containing GZMB and Perforin-1 (PRF1), with PRF1 inducing uptake of GZMB via membrane pores and GZMB, inducing apoptosis by cleaving and activating caspases-3, -7, -8 and -10 and BID [53]. That intrinsic apoptosis may take place in CD is suggested by findings of decreased expression of BCL-2 and increased expression of cytoplasmic tumor protein p53 in the crypts of duodenal tissue from untreated CD patients [54]. Moreover, mRNA levels for BAK (but not BAX) are increased in the duodenum of CD patients, while BAK protein is upregulated in the intestinal epithelium of untreated CD patients and its expression correlates with IFNγ levels [55]. By promoting upregulation of BAK and downregulation of BCL-2, IFNγ has a pro-apoptotic role in epithelial cells in CD [56].

As noted above, an increased number of gliadin-specific, HLA-A2 restricted CTLs has been found in the lamina propria during CD [13], and CTLs of this kind have been shown to cause apoptosis in a model of intestinal epithelial cells in vitro. In addition, an increase in apoptotic enterocytes has been found to correlate with higher cytotoxic activity of IELs in CD and may be associated with higher levels of type I IFNs [21].

### 2.2. Extrinsic Apoptosis

The extrinsic apoptotic pathway is initiated by receptors containing death domains (DD), specifically the TNF family receptors TNFR1, TNFRS10A/B, death receptor 3 (DR3, known as TLIA), and CD95 (also known as FAS). Signals from outside the cells, TNFα, TRAIL, CD95-Ligand (known as FAS-L), bind to these receptors and trigger the assembly of several intracellular multi-protein complexes which ultimately leads to the activation of caspase-8 and caspase-3. In the case of TNFR1, an initial complex named “Complex I”, comprises the adaptor protein TRADD (TNFRSF1A associated via DD) and several additional proteins (TRAF2, TRAF6, c-IAP1, c-IAP2, RIPK1, LUBAC, SHARPIN, HOIL-1 and HOIP). Ligand binding of FAS, DR3, induces the complex named DISC (death-inducing signal complex), made up by the adaptor protein FAS associated via DD (FADD), caspase-8 or 10 (depending on the cell type) and cFLIP (CASP8 and FADD-like apoptosis regulator). These signaling pathways are regulated by the anti-apoptotic factors such as the long isoform of cFLIP (cFLIP-L) and c-IAPs, the trimerization level of the death receptors, as well as by post translational modification of the protein complexes [57]. Caspase-8 is activated by these pathways via autoproteolysis, leading to the generation of cleaved caspase-8 (CC8), which then activates caspase-3, either directly, or indirectly by cleaving BID and promoting MOMP [57].

Activated CTL and NK cells can trigger the extrinsic pathway of apoptosis via the expression of CD95L and TRAIL [58,59,60]. Furthermore, increased expression of CD95 and CD95L has been found in epithelial cells of untreated CD patients [61] and this can be upregulated further by treatment of duodenal biopsies from CD patients with gliadin peptides [62]. In addition, type I and II IFNs may play a role in sensitizing epithelial cells to the extrinsic apoptotic pathway by inducing the expression of both the death receptors and their ligands [63,64].

### 2.3. Common Executioner Pathway of Apoptosis

Key molecules in the apoptosis process can be detected in the small intestine of untreated CD patients, as well as in a mouse model of gliadin-driven innate immunity. Activation of capase-3 is a key event in the common pathway of apoptosis [57] and increased levels of cleaved caspase-3 (CC3) have been found in epithelial cells from intestinal biopsies of untreated CD patients [54]. Recently, we have confirmed these findings and have found that levels of activated caspases-8 and -3 are also increased in newly diagnosed CD patients [65]. In addition, M30, which marks a CC3-degraded form of cytokeratin-18, correlates with CC3^+^ epithelial cells and both M30 and CC3 co-stain with Terminal deoxynucleotidyl transferase dUTP nick end labeling (TUNEL)^+^ apoptotic epithelial cells at the villus tip in CD [55,61,66]. Increased numbers of M30^+^ epithelial cells have also been observed after incubating duodenal biopsies of CD patients with type I IFNs [21].

Studies by our group have shown that oral administration of the innate active p31-43 peptide of gliadin induces upregulation of IFNγ, and increases cell death in the small intestine of wild-type mice, paralleled by an increase in the proapoptotic Bax:Bcl-2 ratio at mRNA level [67]. In parallel, administration of p31-43 leads to increased numbers of TUNEL^+^ cells and increased expression of CC3 by both epithelial and lamina propria cells [42].

## 3. Non-Apoptotic Forms of Programmed Cell Death and Their Implications for CD

Other programed cell death mechanisms have been described in recent years which have necrosis-like phenotypes and lead to the release of inflammatory mediators [1] such as IL-1β, IL-18, and alarmins, even before cell death occurs [68,69]. Some of these non-apoptotic PCD pathways are potentially involved in the pathogenesis of CD.

### 3.1. Pyroptosis

Pyroptosis is a lytic form of programmed cell which leads to rapid clearance of damaged cells during infection. Pyroptotic cells exhibit cell swelling and membrane blebbing, associated with the formation of pores in the cell membrane induced by polymerization of N-terminal fragments of Gasdermin-D (GSDMD). These processes are triggered by a canonical pathway involving the activation of inflammasomes, or by a non-canonical pathway driven by the activation of caspases-4 or -5 (caspase-11 in mouse) [1].

Inflammasomes are a cytosolic multiprotein complex comprising either nod-like receptors (NLRs) (NLR family pyrin domain containing proteins NLRP1, 2, 3, 6, and NLRC4), or non-NLRs (absent in melanoma 2 “AIM2”, and IFNγ-inducible protein 16 “IFI16”) [1,70]. Upon activation, these proteins oligomerize with the apoptosis-associated speck-like protein containing a CARD (ASC), which acts as an adaptor protein that binds and induces autoactivation of caspase-1. When activated, caspase-1 cleaves immature pro-IL-1β and pro-IL-18 into their active forms. In addition, active caspase-1 cleaves GSDMD, producing N-terminal fragments which multimerize and bind to the inner cell membrane, leading to the formation of ~20 nm pores that allow the release of mature IL-1β and IL-18, as well as alarmins such as IL-33, IL-1α and HMGB1 [1,70,71]. GSDMD can also be cleaved by a caspases-4/5 (caspase-11 in mouse)-dependent mechanism in the presence of cytosolic LPS. In this case, caspase-4/5 (or caspase-11) oligomerize and auto-activate, leading to the cleavage of IL-1β, IL-18 and GSDMD [1,72]. There is a balance between pore formation and cell membrane repair, and if the number of pores exceeds the capacity of repair, the cell will die by the process of pyroptosis [1].

Activation of the inflammasome requires two signals. First, a “priming signal” such as TLR ligands, IFNs or alarmins are needed to activate the intracellular NF-kB pathway. The second signal can be produced by a broad arrange of stimuli, whose nature depends on the sensor which has been primed (i.e., members of NLR family, AIM2 or IFI16). These can include MAMPs derived from pathogens, particulate materials (silica and asbestos) or DAMPs such as extracellular ATP, uric acid, and cholesterol crystals, or it may involve additional cellular events, including K^+^ efflux, mitochondrial damage and ROS generation and lysosomal rupture [70,73,74]. Higher levels of ROS [75,76,77], DAMPs such as IL-33 [78] and HMGB1 [79], have been found in increased in CD patients. Interestingly, HMGB1 can promote the activation of pyroptosis by enhancing the delivery of LPS to caspases-4/5 or -11 in the cell cytosol [72].

Components of the local inflammatory response in CD, such as type I and II IFNs can induce the expression of pyroptosis-associated caspase-4 via Interferon Regulatory Factor 1 (IRF1) activation [80], and of the AIM2 inflammasome and IL-18 production via STAT1 [81]. IL-17A, a cytokine upregulated in some CD cases can also induce the expression of NLRP2 and caspase-5 [80].

That these pathways may be participating in CD is suggested by the findings that IL-18 production is increased in small intestinal crypt cells in CD [82] and that there are high levels of circulating IL-18 in untreated CD patients [83]. IL-1β is also found in the supernatants of Peripheral blood mononuclear cells (PBMC) from CD patients upon pepsin-trypsin-treated gliadins exposure [84] and serum levels of IL-1β fall in CD patients after 1 year on a GFD [85].

Similarly, studies using PBMC from CD patients treated with pepsin digested gliadins have shown higher production of IL-1β and IL-18 in cells from CD patients compared with healthy controls. Release of these cytokines was inhibited by blocking K^+^ efflux, suggesting the role of inflammasome in response to digested gliadins in PBMC from CD patients [86].

In vitro work shows that IFNα stimulates caspase-dependent inflammasome activation and IL-18 production in duodenal biopsies from CD patients in vitro, leading to IL-18 dependent IFNγ production [21]. IFNγ itself has also been shown to induce increased expression of NLRP6, and caspases-1 and -5 in enterocytes isolated from CD patients [80].

As noted above, the p31-43 gliadin peptide induces type I IFN production and local inflammation in the small intestine of wild-type mice when given orally, and we have shown that the resulting pathology is dependent on NLRP3 signaling, leading to activation of caspase-1 and IL-1β [42]. Oligomerization of the p31-43 peptide may be responsible for providing signal 2 for inflammasome activation in this context [40].

Interestingly, a group of CD patients share an SNP (rs 12150220 A/A) located in the coding region of NLRP1. This SNP is also found in other inflammatory conditions and is associated with increased IL-18 levels in serum, perhaps accounting for the increased levels of IL-18 seen in some CD patients [87]. A SNP in the IL-1β gene (rs16944 C > T, also known as IL1B -511T) has also been found to be associated with osteopenia/osteoporosis and lower mineral density in CD patients [88]. This SNP is thought to influence the expression levels of IL-1β and is also associated with higher risk of other chronic inflammatory conditions such as Alzheimer’s disease [89], gastric cancer [90], keratoconus [91] and Grave’s disease [92].

Thus, there is evidence in both humans and mice that inflammasome activation and pyroptosis may occur in the small intestine in CD (summarized in Figure 2). As well as potentially explaining some of the epithelial cell death, this mechanism may contribute to other aspects of CD pathogenesis, including the increased production and release of IL-18 and IL-1β, both of which can produce inflammation directly and can activate other immune cells, including Th1, Th17 and CTLs [93,94,95]. IL-1β can also induce expression of a FOXP3 splice variant (FOXP3Δ2Δ7) which is associated with poor regulatory T cell function [96,97] and is associated with CD [98]. The alarmins HMGB1 [79], IL-1α [85] and IL-33 [78,99] are also released during pyroptosis and have been associated with CD, where they may play a number of roles (see below).

### 3.2. Necroptosis

Necroptosis is a necrotic PCD mechanism, which involves the phosphorylation and activation of the membrane pore protein mixed lineage kinase domain-like pseudokinase (MLKL) by activated receptor interacting serine/threonine kinase 3 (RIPK3) [1]. RIPK3 phosphorylation and subsequent activation can be triggered by a broad range of stimuli, including the death receptor ligands that trigger extrinsic apoptosis (TNFα, CD95L, TRAIL). Their ability to induce necroptosis is dependent on apoptosis being blocked, for instance by the presence of caspase-8 inhibitors, or when its activation is prevented by inhibition of the adaptor protein FADD [57,100,101]. When enzymatic activity of caspase-8 is deficient, phospho-RIPK1 binds RIPK3, producing an intracellular protein complex (the necrosome) that recruits and activates MLKL. This leads to the vesicular transport of phospho-MLKL (pMLKL) to the plasma membrane, where it binds to the inner cellular membrane and forms a pore [102]. When MLKL activation exceeds the membrane repair capacity, the loss of membrane integrity triggers necrosis and the release of alarmins such as ATP, HMGB1, IL-1α and IL-33. RIPK1-independent necroptosis can also occur when RIPK3 is activated by viral dsRNA-mediated activation of the TLR3-TRIM pathway [103] and by activation of Z-DNA Binding Protein 1 (ZBP1) [104,105].

Untreated CD patients showed significantly higher mRNA expression of RIPK3, ZBP1 and MLKL, suggesting that necroptosis may account in part for the increased cell death found in active CD [65]. Furthermore, oral administration of gliadin p31-43 led to increased cell death and RIPK3 expression in small intestinal epithelial and lamina propria cells in wild-type mice [42]. Necroptosis was also observed in Paneth cells in inflamed ileal tissues from patients with Crohn’s disease, and experiments in mice demonstrated that Paneth cells follow a necroptosis pathway, via IFNs/STAT1 and MLKL, controlled by caspase-8 [106].

After necrosome formation, pMLKL is transported inside vesicles to the plasma membrane escorted by proteins involved in the structure of tight junctions, and among them is Zonula occludens-1 (ZO-1), which can inhibit pore formation by pMLKL [102]. Deposition of MLKL pores and necroptosis in enterocytes due to downregulation of ZO-1 in untreated CD is thought to contribute to the loosening of tight junctions and loss of the integrity of the epithelial barrier [107,108].

Although some of the pro-necroptotic factors and positive modulators of necroptosis have been found in CD (summarized in Figure 3, Top Image), further investigation is needed to determine the role of necroptosis in this pathology.

### 3.3. Ferroptosis

Ferroptosis is a non-apoptotic-like PCD, with exclusive biochemical and morphological changes [109] where iron ions catalyze oxidative reactions on poly-unsaturated fatty acids (PUFA) mainly found in the mitochondrial and plasma membrane. The phenotype of these cells differs from classical necrotic or apoptotic cells with a characteristic dysmorphic permeabilized outer membrane of mitochondria and damaged plasma membrane occurring due to excessive oxidation of membrane lipids [1,110,111,112]. This new form of PCD leads to the release of DAMPs into the extracellular space (summarized in Figure 3, Bottom Image) [113]. Ferroptosis is induced upon inhibition of the phospholipid peroxidase and oxidoreductase glutathione peroxidase 4 (GPX4) enzyme and reduction of GPX4 substrate, glutathione (GSH) [1,114]. GPX4 is a selenium-enzyme which is part of the major protective mechanism against lipid peroxidation, it uses GSH to reduce H_2_O_2_, organic and lipid peroxides. Thus, GPX4 protects cell membranes from the hazards of oxidants molecules, and as a result, the inhibition of GPX4 increases lipid peroxidation. Ferroptosis can also occur secondary to an increase in the intracellular free iron pool (iron overload) which leads to increased H_2_O_2_ levels via a series of Fenton reactions [1,69,110].

Increases of ROS, total lipid hydro-peroxides (LOOH) and reduced antioxidant processes have been reported in the peripheral blood and duodenal mucosa of CD patients and these changes are reverted by a GFD [76,77]. In parallel, ROS and nitric oxide production are increased in circulating erythrocytes and the small intestine of untreated CD patients, together with reduced levels of GSH, the principal substrate of GPX4 [75,76,77]. Increases in other markers of oxidation, including catalase, superoxide dismutase, myeloperoxidase and DNA instability have also been detected in peripheral blood of untreated CD patients [115]. A further factor which could lead to reduced GPX4 activity in untreated CD may be the deficiency in selenium uptake which occurs in these patients [116]. Since IFNγ and p53 inhibit the expression of membrane cellular cysteine transporters (SLC3A2 and SLC7A11) [117,118], it may potentiate the induction of ferroptosis in CD.

## 4. Alarmins and DAMPs

DAMPs (damage-associated molecular patterns) are a set of highly immunogenic molecules which are associated with cell damage, including Type I IFNs, IL-15, which have been linked to CD [17,69,119,120]. Alarmins, a subgroup of DAMPs, are endogenous molecules released upon cell damage (during spontaneous necrosis, necrosis PCD or necrosis-like PCD) that trigger a response on different immune and non-immune cells [2,121]. The release of DAMPs may potentiate and expand the inflammatory process even at distant sites [122] (summarized in Figure 4). Here, we are going to describe HMGB1 and IL-33, which are released under necrotic cell death and have been already associated with CD.

### 4.1. HMGB1

The high-mobility group box 1 protein (HMGB1) is normally found in the nucleus of a variety of cells where it acts in different DNA repair mechanisms and increases the affinity of several transcription factors to its cognate DNA sequences [123]. However, under certain inflammatory situations, HMGB1 can be relocated in the cytoplasm and then secreted by lysosomal traffic or during necrosis or necrosis-like PCDs [124,125]. Free HMGB1 binds to TLR4 and induces inflammatory reactions depending in the redox state of its cysteine residues [126]. Interestingly, HMGB1 can also bind to LPS and serves as a carrier to trigger TLR4 activation. On the other hand, the receptor for advanced glycation end products (RAGE) is a HMGB1 receptor, which triggers activation of endothelial cell and smooth muscle cell proliferation [127,128]. HMGB1 can also have chemotactic activity, and immune cell activation through TLR4, RAGE and other immune-related receptors [129]. HMGB1 can activate pyroptosis by its capacity to permeabilize endo-lysosomal membranes at low pH and releasing LPS and cathepsin-B into the cytoplasm. This process leads to the activation of several inflammasome sensors by cathepsin-B or caspases-4 or -5 (caspase-11 in mice) by direct binding with LPS (summarized in Figure 2 and Figure 4) [72,130].

Increased levels of HMGB1 have been found in the serum and feces of untreated CD patients [131,132,133], as well as in the blood during autoimmune disorders associated with CD, such as type I diabetes mellitus, Sjogren’s syndrome, and autoimmune thyroiditis [134,135,136,137,138,139]. Moreover, HMGB1 was found to be associated with an increased capacity of dendritic cells to promote a pro-Th1 phenotype in T cells during antigen presentation, to expand CTLs cell populations, and to promote M1 polarization of macrophages (Figure 4) [140,141,142], all this highlights HMGB1’s role in driving inflammation. Thus, these findings suggest that HMGB1 could play a role in the inflammatory response in CD, and this needs further investigation.

### 4.2. IL-33

IL-33 is a member of the IL-1 family which is normally located in the nucleus of mesenchymal and epithelial cells, but it can be actively released from viable cells or passively from cells undergoing inflammatory PCD; this mechanism is enhanced if IL-33 is translocated to the cytoplasm by unknown mechanisms [143,144,145]. Normally, IL-33 release is prevented during apoptosis by cleavage of IL-33 into inactive fragments by caspases-3 and -7 [146]. By acting via its specific receptor IL-33Rα (also known as ST2L) [147], has a variety of effects on immune cells (summarized in Figure 4) and is known particularly for its role in allergy and parasite infections because of its ability to activate ILC2, Th2 and mast cells [148,149,150]. However, IL-33 can also promote Th1 and importantly, pro-cytotoxic CD8^+^ T cell activity during viral infections and immune responses to tumors [151,152], and can maintain survival of regulatory T cells [153]. Notably, IL-33 also has an effect on intestinal epithelial cells, inducing their proliferation and stimulating secretion of protective antimicrobial peptides from enterocytes [154,155]. We and others have shown an increase in IL-33 expression in the serum of untreated CD patients [78,99]. Moreover, we have found a large number of different cells associated with microvasculature (characterized by the expression of SMA, CD31 and CD90), with nuclear IL-33 location and others with a cytoplasmic accumulation of this cytokine. Western blot analysis of duodenal mucosa from CD patients also revealed increased levels of the 18–21 kDa sized fragments of IL-33 [78] that are produced by the action of enzymes released by activated neutrophils and mast cells and which have increased affinity for the IL-33R [156,157,158]. In parallel, increased numbers of CD8^+^IL-33R^+^ cells are found in duodenal mucosa in untreated CD, suggesting that these free IL-33 fragments could potentiate the cytotoxic actions of CTLs in CD patients [78]. Additionally, IL1RL1, the gene that codes for ST2, has been linked with CD disease SNP (rs1420106) [159]. Moreover, isolated gluten-specific T cells clones overexpressed IL1RL1 after proper gluten challenge [159]. These findings highlight a potential role of IL-33 axis in CD patients.

## 5. Potential Interplay between Different PCDs

As we have discussed, untreated CD patients have increased numbers of cells dying via apoptosis and by other pro-inflammatory PCD pathways.

Cell death pathways also are interconnected and influence each other. For instance, there is evidence that lipid peroxidation changes associated with ferroptosis may occur in CD [75,76,77,115] and it has been shown that these mediators can induce apoptosis in the presence of a competent antioxidant system (GSH and thioredoxin systems). However, when the antioxidant system is deficient, both apoptosis and pro-inflammatory PCDs may occur because oxidative conditions activate factors such as MLKL, RIPK1/3, NLRP3, caspase-1 and GSDMD. In contrast, the activation of pro-apoptotic caspases-3 and -7 requires effective antioxidant mechanisms [160]. Thus, CD patients that have a deficit in antioxidant capacity [76,77] may develop a greater activation of pro-inflammatory PCDs. On the other hand, CD patients have been shown to have enhanced IFNγ-mediated induction of thioredoxin (Trx1)-dependent antioxidant systems [161], and Trx1 is thought to support the activation of TG2 activity [162]. Further studies on the oxidative mechanism are needed to assess its relevance in modulating PCDs in CD patients.

The co-existence of multiple cell death pathways has been found in other inflammatory conditions such as Crohn’s disease [106], and in an enteropathy experimental model induced by a single dose of p31-43 [42]. This phenomenon has been referred to as PANoptosis and it is defined as the outcome of evolutionary conserved, interrelated pathways of cell death, which leads to different outcomes (apoptosis, necroptosis and pyroptosis) [163]. Furthermore, it is suggested that this process involves a unique pathway controlled by a multiprotein complex called the “PANoptosome” based on ZBP1. Inflammatory responses driven by activation of the PANoptosome have been postulated in neurodegenerative diseases, cancer, infection-driven inflammation, joint inflammation, and metabolic inflammation [163].

In turn, cytokines and alarmins may modulate PCDs pathways. IFNs are potent modulators of different PCDs, by stimulating the expression of proapoptotic proteins such as BAK, NOXA, caspase-8, and death receptors or death ligands (TNFα, CD95L), while promoting also anti-apoptotic proteins such as cFLIP [64,164]. IFNs may also control the expression of pyroptotic proteins such as caspases-4 and -5, caspase-1 and AIM2, NLRP6, NLRC5 [80,81]. Additionally, IFNs trigger necroptosis by the STAT1 dependent pathway or by increasing ZBP1 and MLKL [104,106,165]. As discussed above, the inflammatory processes triggered in CD may enhance cell death mechanisms of different kinds, leading to the release of HMGB1, IL-1β and IL-33, which expand the inflammation and the induction of further proinflammatory cell death (summarized in Figure 4). Nutritional deficiencies due to mucosal malabsorption and inflammatory process are associated with a broad range of extraintestinal conditions (dermatological, endocrine, and reproductive disorders, neurological and psychiatric conditions, musculoskeletal manifestations, among others) in CD patients [166]. Therefore, inflammation beyond the small intestine deserves further investigation, in order to gain insight into the mechanisms playing a role in proximal small intestines from CD patients, which may be involved in driving systemic disorders.

## 6. Conclusions

The past few years have seen an expansion in the knowledge about programmed cell death, not only in a detailed description of the molecular mechanisms, but also in the discovery of new pathways occurring under specific conditions. This broad field has an immense impact in health and disease. Particularly, different programmed cell death pathways may occur in enterocytes and *lamina propria* cells from the small intestine. In addition to silent apoptosis driven by cytotoxic lymphocytes, proinflammatory processes such as pyroptosis, ferroptosis and necroptosis can also be induced. The alarmins and other pro-inflammatory mediators released by these PCD pathways may play a role in expanding the local inflammatory reaction and, by sustaining the T cell-driven tissue damage, help to the loss of tolerance to gluten-derived peptides in the small intestine of CD patients. Release of inflammatory mediators by proinflammatory PCD may have also a role in potentiating and expanding the tissue damage locally and at distant sites, which may favor the development of chronic inflammatory processes and autoimmunity in susceptible individuals. Control of these PCD pathways may have a therapeutic benefit. However, further efforts will need to validate findings from animal models to human diseases.

## Figures and Tables

**Figure 1 ijms-22-07426-f001:**
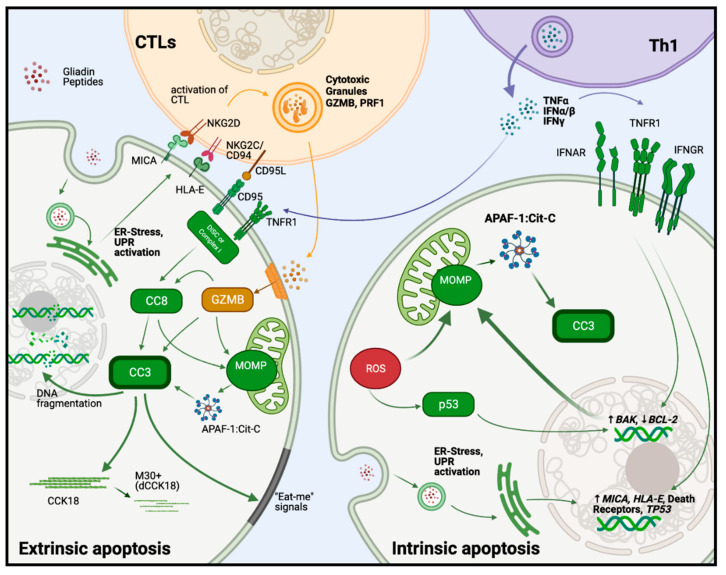
Extrinsic and intrinsic mechanisms of apoptosis playing a role in celiac disease mucosae. On the left side of the picture, there is the scheme of the extrinsic apoptosis mechanism triggered by a CTL. As shown in the picture, in the celiac disease mucosae, the CTLs could trigger apoptosis directly by recognizing stressed-related membrane proteins (MICA, HLA-E) with specific NK receptors (NKG2D, CD94:NKG2C). This event promotes the degranulation of CTLs which release Granzyme B (GZMB) and Perforin-1 (PRF1) in the extracellular medium. PRF1 leads to GZMB entry into the cell cytoplasm which leads to the activation of executioner caspase-3, caspase-8, and MOMP formation. CTLs could also engage death receptor ligands (CD95, TNFα, etc.) to trigger the caspase-8 activation which activates caspase-3. Ultimately, cleaved caspase-3 (CC3) activates the DNA and cytoskeleton protein fragmentation (i.e., dCCK18) and promotes the phosphatidylserine translocation into the outer cell membrane. The last event is recognized as an “eat-me” signal by local phagocytes, which eliminated the apoptotic bodies. The right side of the picture describes different intrinsic apoptotic mechanisms playing a part in celiac disease pathology. Inflammatory cytokines related to CD (IFNγ, TNFα, IFNs type I) and specific gliadin peptides (i.e., p31-43) induce a proapoptotic balance between the BCL-2 family of proteins by increasing pro-apoptotic BAK and repressing BCL-2. This renders a cell prone to induce apoptosis when new stressing conditions appear. Potential stressors are the unfolded protein response (UPR), stressed vesicle traffic (i.e., associated with p31-43 toxicity), and in particular the production of ROS. These stressors induce p53 activation and feed a positive loop back to a proapoptotic BCL-2 family protein balance. If these stressful conditions worsen, the MOMP is formed releasing cytochrome-c into the cytoplasm. This event activates APAF-1 and induces the apoptosome formation which subsequently leads to caspase-3 activation initiating the apoptosis execution phase.

**Figure 2 ijms-22-07426-f002:**
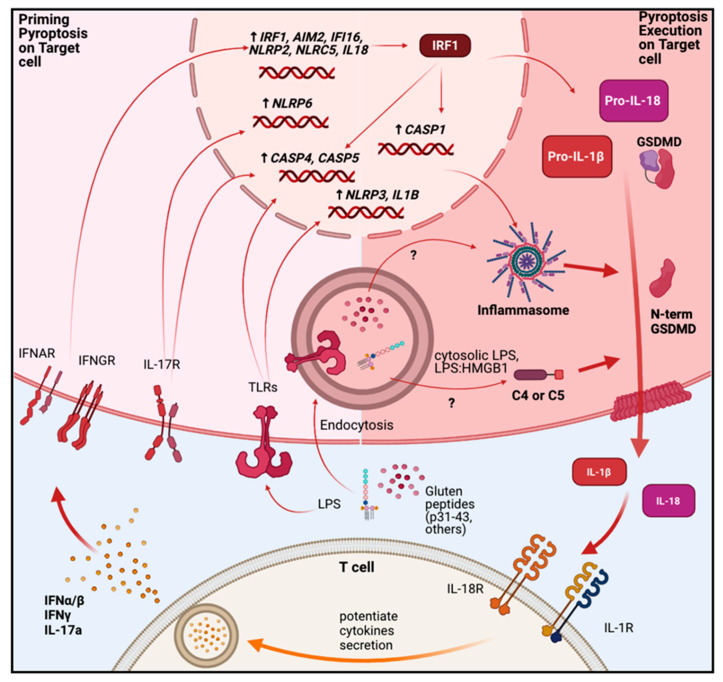
Pyroptosis a new potential PCD mechanism feeding the inflammatory response in celiac disease patients. IFNs which are upregulated in untreated CD induce the expression of inflammasome sensor components such as AIM2, IFI16, NRLP2, NLRC5, transcription factor IRF1, and cytokine IL-18. At the same time, IRF1, a key factor in CD pathology, can enhance the expression of other inflammasome components whose expression is induced by IL17A and TLRs (caspases-4 and -5, NLRP3 and cytokine IL-1β). This process sensitizes the targeted cells to different inflammasome activating agents. Among them, gliadin peptides, such as p31-43, can activate NLRP3 inflammasome (i.e., p31-43). Additionally, the activation of caspases-4 and -5 could be triggered by potential intracellular LPS release, not currently studied in CD. Ultimately, these events activate caspases-1, -4, and -5 which process pro-IL-1β and pro-IL-18 into their mature bioactive forms. Importantly, during this process, Gasdermin-D is cleaved into N-terminal fragments (GSDMD N-Term) which form oligomers in the cell membrane leading to the release of IL-1β and IL-18 and eventually to pyroptotic cell death. The release of these proteins, as well as other alarmins, activates the local immune cells feeding the local inflammatory response by interacting with its cell receptors (IL-18R and IL-1R).

**Figure 3 ijms-22-07426-f003:**
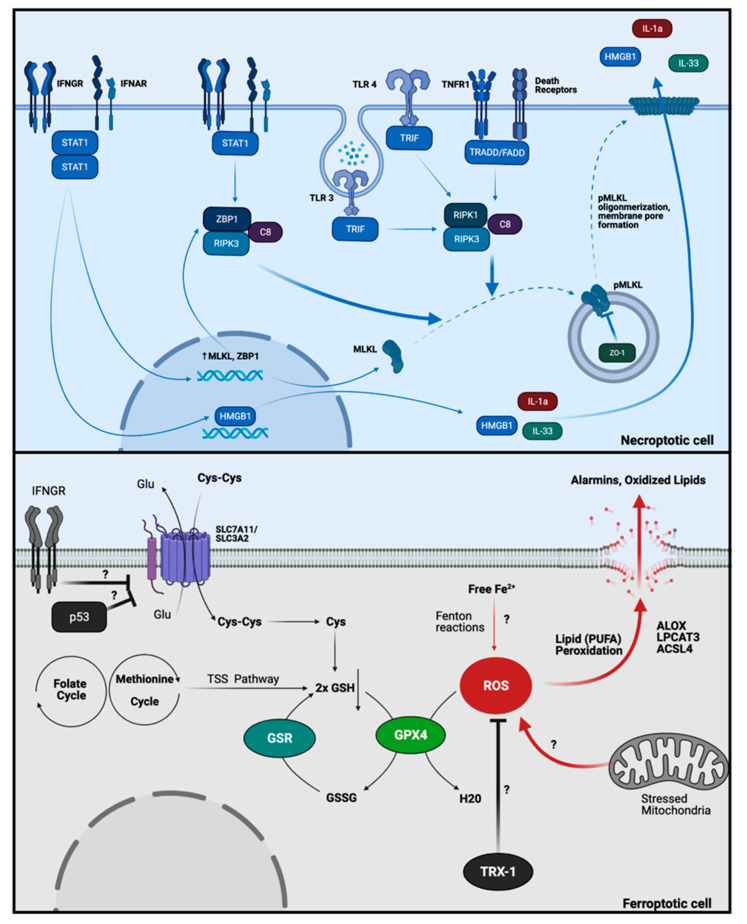
Other potential PCD mechanisms in CD pathology. (**Top Image**): Necroptosis. IFNs are a hallmark of CD patients, and they can stimulate the expression of necroptotic factors such as MLKL and ZBP1 and stimulate the cytoplasmic localization of HMGB1. At the same time, ZBP1 is fundamental for the direct activation of necroptosis by IFNs when caspase-8 is inhibited, by an IFNs/STAT1/ZBP1/RIPK3 mechanism. Additionally, activation of TLRs/TRIF, death receptors (TNFR1, CD95)/TRADD/FADD can induce necroptosis by activating RIPK3 in presence of caspase-8 inhibition. RIPK3 activation leads to MLKL phosphorylation (pMLKL) and migration to the cell membrane in vesicle co-transported with a regulatory factor, ZO-1. As ZO-1 is downregulated in epithelial cells of untreated CD patients, it may indicate a pro-necroptotic phenotype of this cell type. When pMLKL monomers are oligomerized in the plasma membrane, this creates a pore, which leads to necroptosis cell death and the alarmins (IL-1α, IL-33, HMGB1) release. (**Bottom Image**): Ferroptosis. CD patients show a dramatic increase in ROS production and oxidative stress markers which could be related to a new form of inflammatory PCD called ferroptosis. This PCD could be triggered by inhibiting GPX4 through a reduction of GSH. Both facts have been found in untreated CD patients’ cells. Additionally, IFNγ and p53, both factors related to CD pathology, can reduce the transport of cystine (Cys-Cys) a critical precursor of GSH biosynthesis by inhibiting its membrane transporter (SLC3A2 and SLC7A11). Moreover, the GPX4 downregulation could be a response to the ROS increase if other reductive systems failed (i.e., TRX1). The massive increase in ROS could be triggered by stressed organelles such as mitochondria and the free iron pool. Ultimately, the reduced activity of GPX4 leads to ALOXs, LPCAT3, and ACSL4 increase in membrane PUFA peroxidation and cell death, with the release of alarmins and inflammatory oxidized lipids.

**Figure 4 ijms-22-07426-f004:**
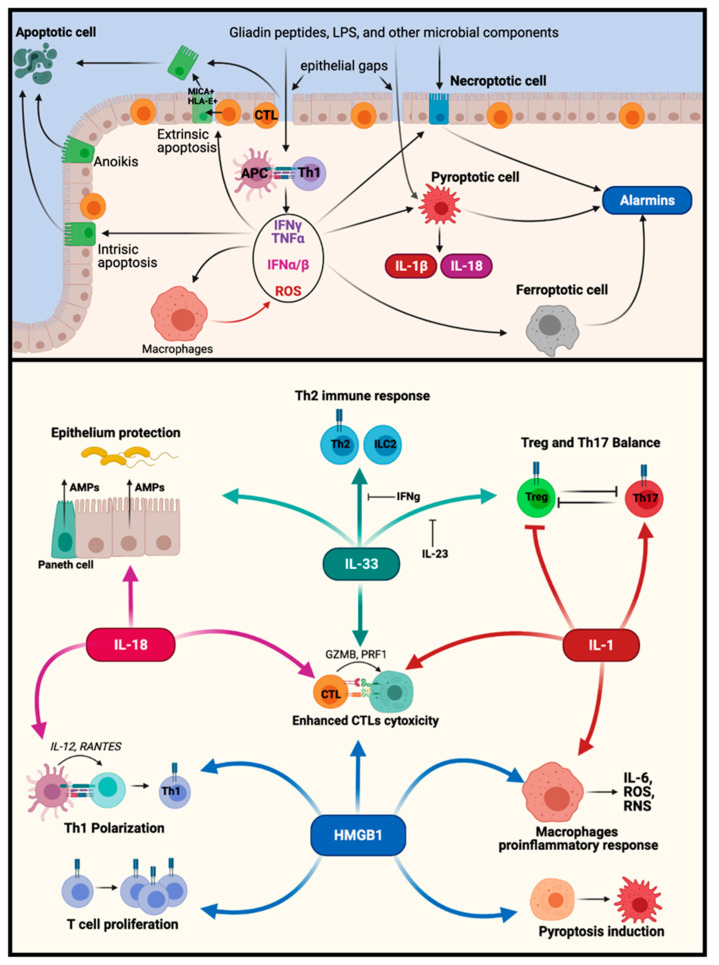
A new complex hypothetic scenario for CD pathology. (**Top Image**): Coexistence of different PCD in CD mucosae. Conclusions. Extrinsic apoptosis is primarily triggered by NK-like CTLs in the epithelium and stressed target cells (MICA^+^, HLA-E^+^) in the epithelial compartment. At the same time, inflammatory cytokines released by Th1, CTLs (IFNγ, TNFα), activated dendritic cells and other antigen presenting cells (APC) (Type I IFNs) and activated macrophages (ROS) sensitize different cells, especially epithelial cells, to trigger intrinsic apoptosis. After induction of apoptosis, these cells are detached from the epithelium and are reduced to apoptotic bodies in the lumen. In the end, the excessive death is answered with an increased proliferation rate of transit-amplifying cells. However, the presence of these inflammatory cytokines (IFNγ, TNFα) deteriorates the epithelium capacity to efficiently close the gaps left by apoptotic cells. This may create a series of “epithelial gaps” where microbial components (MAMPs) and gliadin peptides could be introduced in the lumen, triggering a new wave of the inflammatory response by gluten-specific T cells. As previously described, this scenario is appropriate to trigger pyroptosis and necroptosis on sensitized cells. In the end, all the new PCDs could trigger the release of inflammatory factors, such as alarmins and IL-1β and IL-18, which may feed the inflammatory process. (**Bottom Image**): Potential effect of alarmins and IL-1β/IL-18 in CD mucosae. The different immunogenic factors released from necrotic PCD induce different immune and cell responses. IL-33 and IL-18 have a protective effect on epithelium by increasing epithelial proliferation and AMPs released into the lumen. At the same time, IL-33 has a pro-Th2 and pro-Treg action, which is specifically inhibited by inflammatory cytokines such as IFNg and IL-23. Additionally, IL-1α and IL-1β (IL-1) inhibit Treg phenotypes, potentiating the Th17 polarization and the inflammatory response. Additionally, pro-cytotoxic capacity of CTLs can be enhanced by the presence of IL-33, IL-1, HMGB1 but especially IL-18. Furthermore, IL-1 and HMGB1 enhance the inflammatory capacity of macrophages, but HMGB1 may lead to an eventual pyroptotic cell death acting as a lysosomal LPS carrier. Additionally, HMGB1 promotes T cell proliferation on different T cell populations, and similarly to IL-18, potentiates Th1 polarization during antigen presentation. All this information suggests an exciting future in the study of CD pathology, and the very likely involvement of different PCD mechanisms in the overall pathogeny process and new animal models of CD-triggering events.

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
