# Peer review of "Programmed Cell Death in the Small Intestine: Implications for the Pathogenesis of Celiac Disease"

_ijms, 2021, doi:10.3390/ijms22147426_

Round 1

Reviewer 1 Report

In this manuscript, Perez et al. summarized the recent literature addressing the role of programmed cell death pathways in the small intestine, describing how these mechanisms may contribute to CD and discussing their potential implications. The English reads well. Moreover, I appreciate the figures the authors created which help readers to understand the mechanisms clearly.

Several suggestions I would like to rise are as follows,

  1. Please reduce the numbers of keywords. Just pick up the most important ones and insert them.
  2. Please modify the conclusion section. In this section, I would like to know more about where the further investigations need to go.
  3. Please reduce the numbers of references. In the context, it is just necessary to cite the most important references instead of listing all the relative papers.

Author Response

Reviewer 1

In this manuscript, Perez et al. summarized the recent literature addressing the role of programmed cell death pathways in the small intestine, describing how these mechanisms may contribute to CD and discussing their potential implications. The English reads well. Moreover, I appreciate the figures the authors created which help readers to understand the mechanisms clearly.

Several suggestions I would like to rise are as follows,

  1. Please reduce the numbers of keywords. Just pick up the most important ones and insert them.
  2. Please modify the conclusion section. In this section, I would like to know more about where the further investigations need to go.
  3. Please reduce the numbers of references. In the context, it is just necessary to cite the most important references instead of listing all the relative papers.

Answer

We thank to reviewer 1 for his/her positive consideration of our manuscript

  1. Four keywords were deleted
  2. Conclusion section was modified
  3. Seventeen references were deleted according with this list:

Lane 319: Wong 2013(NK), O’Sullivan 2006., Mailer 2015

Lane 347:104 Vercammen 1998

Lane 381: Linkermann 2014; Proneth 2019

Lane 389: Proneth 2019.

Lane 423: Sharma 2015.

Lane 435: Travers 2003; Rowell 2012; Brezniceanu 2003;

Lane 437: Gardella 2002; Ranzato 2015;

Lane 454: Rovere-Querini 2004; Andersson 2018

Lane 469: Molofsky 2015; Ochayon 2020.

Reviewer 2 Report

In this review Perez and Coll. summarized recent literature related to different forms of pro-inflammatory programmed cell death (PCD) such as pyroptosis, ferroptosis  and necroptosis of which they described molecular mechanisms and  implication in pathogenesis of small intestine inflammatory diseases.

In particular, they focused on the role of PCD in pathogenesis of Celiac Disease (CD) and, although the literature about this argument is scarce or almost inexistent, they suggested  a possible scenario aimed at demonstrating a potential involvement of these forms of cell death in CD. 

The paper is interesting and well written and, albeit a little speculative, however, able to stimulate new studies in the field.

I only suggest to reduce figure legends, in particular the fourth, and perform a new paragraph, before conclusion, illustrating the hypothetic scenario for CD pathogenesis in which different forms of PCD participate and interact with inflammatory mediators.

Author Response

Reviewer 2

In this review Perez and Coll. summarized recent literature related to different forms of pro-inflammatory programmed cell death (PCD) such as pyroptosis, ferroptosis  and necroptosis of which they described molecular mechanisms and  implication in pathogenesis of small intestine inflammatory diseases.

In particular, they focused on the role of PCD in pathogenesis of Celiac Disease (CD) and, although the literature about this argument is scarce or almost inexistent, they suggested  a possible scenario aimed at demonstrating a potential involvement of these forms of cell death in CD. 

The paper is interesting and well written and, albeit a little speculative, however, able to stimulate new studies in the field.

I only suggest to reduce figure legends, in particular the fourth, and perform a new paragraph, before conclusion, illustrating the hypothetic scenario for CD pathogenesis in which different forms of PCD participate and interact with inflammatory mediators.

 Answer

We thank to reviewer 2 for his/her positive consideration of our manuscript

Legend of figure 4 was reduced (from 653 words in the original version to 391 in the current one).

As regards the reviewer’s suggestion, Section 5 “Potential interplay between different PCDs“ aimed to describe an integrative view of all information discussed in the review on PCDs and celiac disease and to consider also the potential link with other disorders. We do not think that by  extending the length of this section we will be able to add more insights on this topic